# Control and Signal Acquisition System of Broad-Spectrum Micro-Near-Infrared Spectrometer Based on Dual Single Detector

**DOI:** 10.3390/mi12060696

**Published:** 2021-06-15

**Authors:** Haitao Liu, Shuoran Niu, Ying Zhou, Liwei Yu, Yue Ou, Jiayao Ding

**Affiliations:** 1School of Microelectronics and Communication Engineering, Chongqing University, Chongqing 400044, China; 201812131027@cqu.edu.cn (S.N.); 20193727@cqu.edu.cn (L.Y.); 2Key Laboratory of Fundamental Science of Micro/Nano-Device and System Technology, Chongqing University, Chongqing 400044, China; yzhou@cqu.edu.cn; 3School of Automation, Chongqing University, Chongqing 400044, China; 20193751@cqu.edu.cn (Y.O.); 20193750@cqu.edu.cn (J.D.)

**Keywords:** micro-NIR spectrometer, scanning grating mirror, dual single detector, control, acquisition

## Abstract

Based on the scanning grating mirror developed by us, this paper presents a method for precise control of the scanning grating mirror and high-speed spectrum data acquisition. In addition, a system circuit of the scanning grating mirror control and a spectrum signal acquisition system were designed and manufactured. The final results of the experiment show that the control system successfully allowed the precise control of the swing of the scanning grating mirror and the acquisition system successfully carried out the high-speed acquisition and transmission of the spectrum and angle data. The spectrum detection range of the NIR spectrometer was 80–2532 nm. The overall resolution of the spectrum was better than 12 nm.

## 1. Introduction

The micro-NIR (micro-near-infrared) spectrometer can realize real-time and rapid detection in the field and has the characteristics of convenient carrying, simple operation and wide application for the requirements of the fields of metallurgy, agriculture, medicine, the petroleum industry, the chemical industry, environmental monitoring, aerospace technology, food safety, etc. [1,2,3,4,5,6,7].

At present, many commercialized micro-NIR spectrometer products have the common characteristic of working with a fixed diffraction grating and an InGaAs detector array, such as Ocean optics, United States; Avantes, Netherlands; Hamamatsu, Japan; INSION, Germany; FuXiang, China, etc. [8,9,10]. This configuration inevitably results in high production costs and a large volume, which largely limit the development of this type of spectrometer.

With the rapid development of MOEMS technology, more and more companies and scientific research institutions develop various types of micro-NIR spectrometers based on MOEMS technology, including micro-NIR spectrometers based on FT (Fourier transform) [11,12,13,14,15], micro-NIR spectrometers based on the FP (Fabry–Perot) interferometer [16,17,18], micro-NIR spectrometers based on grating light modulator [19,20,21,22] and micro-NIR spectrometers based on a scanning grating mirror [23,24,25,26,27]. By replacing the traditional fixed diffraction grating with an MOEMS scanning grating mirror in a micro-NIR spectrometer, the cheap InGaAs single detector diode instead of the expensive InGaAs detector array can be used to detect the continuous spectrum. The cost and volume of the miniature spectrometer are greatly reduced. In all of these micro-NIR spectrometers based on a scanning grating mirror, because scanning gratings do not integrate a deflection angle sensor, the photodiode is used to replace the position-sensitive detector (PSD) to acquire the information about the deflection angle of the scanning mirror and the closed-loop control of the scanning mirror, which would enlarge and complicate the whole system. Furthermore, the data acquisition system is bulky and complex, which cannot meet the special application requirements of micro-NIR spectrometers based on the MOEMS scanning grating mirror.

Based on the scanning grating mirror integrated with an angle sensor developed by us [28,29], we present a method of precise control of the scanning grating mirror and high-speed spectrum data acquisition. In addition, the system circuit of the scanning grating mirror control and spectrum signal acquisition was designed and manufactured. The final results of the experiment show that the designed system can successfully achieve the precise control of the scanning grating mirror and obtain spectrum data.

## 2. Principle of System Structure

Figure 1 shows the principle of the system structure of the micro-NIR spectrometer based on the MOEMS scanning grating mirror. The light beam emitted from the NIR light source passes through the entrance slit to the concave mirror and irradiates the surface of the MOEMS scanning grating mirror after collimation; and the two beams of monochromatic light of different wavelengths enter the focusing mirrors 1 and 2; then the two beams pass through the exit slits 1 and 2, and enter the InGaAs single-tube detector; that is, the monochromatic light is reflected through the concave mirror to the exit slit and detected by the InGaAs single detector diode. At the same time, the output angle signal of the integrated angle sensor on the surface of the scanning grating mirror acts as a reference signal for closed-loop feedback control and also as a synchronous trigger signal for spectrum acquisition. Finally, the output of the InGaAs single detector diode and the output of the angle sensor signal are both transmitted to the computer through the USB interface for spectrum reconstruction, display and storage. By designing a reasonable scanning angle, the monochrome light in a certain band can enter the corresponding InGaAs single-tube detector only through a focusing mirror so as to ensure that the two channels do not interfere with each other. 

The spectrum detection range of one channel of a dual detector miniature near-infrared spectrometer is 1600–2500 nm. By designing a reasonable scanning angle, a certain band of monochromatic light passes through only one focusing mirror and then enters the corresponding InGaAs single-tube detector, ensuring that the two channels do not interfere with each other. The dual detector micro-near-infrared spectrometer has a spectrum detection range of 800–1600 nm and the spectrum detection range of the second channel is 1600–2500 nm.

The MOMES scanning grating mirror includes a movable mirror plate, a pair of torsional bars, integrated blazed grating, driving and angle sensing coils, lead wires, electrode pads and a pair of permanent magnets. Figure 2 shows the schematic diagram of the MOEMS scanning grating mirror. The movable mirror plate is attached to the outer fixed frame via a pair of rectangular torsional bars. One side of this device integrates with the blazed grating, the other integrates the driving and angle sensing coils. The driving and angle sensing coils are connected to the electrode pads by lead wires which cross the torsional bars. Moreover, a pair of permanent magnets is assembled, which mainly generates a magnetic field; therefore, the Lorentz force actuates the mirror plate scanning around the beam. The induced electromotive force is generated through the angle sensing coil due to the reciprocating motion of the movable mirror plate in the magnetic fields. Additionally, the integrated blazed grating can diffract and scan the spectra simultaneously. When an excitation signal equal to the first-order natural frequency of the MOEMS scanning grating micro-mirror is applied to the driving coil, the device can revolve around the torsion beam in a resonant state. Subsequently, the MOEMS scanning grating micro-mirror achieves a larger scanning range at a lower driving voltage.

However, in practical engineering applications and experiments, there are some uncertainties and disturbances that affect the stability of the micro mirror, such as device processing and assembly, and some environment temperature, humidity, self-heating, electromagnetic interference and environmental vibration. Therefore, in order to make the MOEMS scanning grating micro-mirror work in a stable state for a long time, our team used a closed-loop control to control the micro-mirror. The control scheme includes an angle sensing module, a feedback control module and a driving module.

The output voltage waveform of the MOEMS scanning grating micro-mirror angle sensor is stable and its noise is small. It has a good linear relationship with the scanning angle. Therefore, the real-time measurement of the scanning angle of the MOEMS scanning grating micro-mirror can still be realized effectively.

However, in the process of its operation, the amplitude of the swing of the MOEMS scanning grating mirror will change because of thermal effects, environmental factors and other factors; the closed-loop feedback control must be added to ensure the amplitude of the swing of the scanning grating mirror constantly. Figure 3 shows the structure of the drive and control module of the scanning grating mirror. The deviation signal is obtained by comparing the reference signal with the obtained detection signal. The deviation signal and voltage signal of the VGA can be controlled by DAC conversion of the deviation signal after PID operation and one can control the drive signal through the variable gain amplifier (VGA). Finally, the size of the driving micro-mirror signal is adjusted in time to realize the position of the micro-mirror and the amplitude of the swing of the scanning grating mirror is constant. The closed-loop feedback control system can suppress or eliminate the deviation in order to make the scanning grating mirror swing constantly.

In the signal acquisition and processing module, the output spectrum signal and the angular sensing signal of the two detectors are collected by the ADC and, after being buffered in the FPGA, the USB interface is sent to the upper computer to realize the reconstruction of the spectrum curve in the upper computer. Figure 4 shows the structure of the signal acquisition system of the micro-NIR spectrometer based on the MOEMS scanning grating micrometer, which includes a C-V (current-voltage) conversion module, a signal preprocessing module, an ADC (analog digital conversion) module, an FPGA (field-programmable gate array) control module and a USB (Universal Serial Bus) interface module. The FPGA module receives the command from the host computer and controls the spectrometer. The InGaAs single detector diode converts the spectrum signal into an electrical signal and transmits to the ADC module. Additionally, the output angle signal of the angle sensor is also transmitted to the ADC module circuit after preprocessing and the ADC module converts all these analog signals into a digital signal. The output signal of the ADC is transmitted into the FPGA module and to the host computer through the USB interface after FPGA processing; the computer performs the reconstruction and display of the spectrum information.

## 3. Circuits and Algorithm Design

According to the principles of the control and detection system of the micro-NIR spectrometer (shown in Figure 3 and Figure 4), the mirror control circuit and the signal acquisition circuit were designed and manufactured.

### 3.1. Mirror Control System (Drive and Closed-Loop Feedback Control System)

The previous study found that there was a good linear relationship between the scanning grating mirror drive signal and the output signal of the angle sensor. The output of angle sensor is a sinusoidal signal. The amplitude of the angular sensing signal is controlled by the sliding-film PID transformation in the FPGA after sampling and AD conversion and the amplitude of the driving signal is controlled to achieve the constant amplitude of the micro-mirror.

#### 3.1.1. The Drive Signal Generation Module Based on DDS

In the design of this micro-NIR spectrometer, the sinusoidal signal required for the micro-mirror to be generated by DDS in the FPGA was adopted. The frequency is obtained and adjustable by the host computer. The principle of DDS signal synthesis is to store the required output waveform in the storage space. Then, the frequency control word is used to control the address accumulation and the frequency-adjustable signal is then generated at the output, as shown in Figure 5.

#### 3.1.2. DAC and Low Pass Filter Circuit

In order to achieve high precision control of the MOEMS scanning grating micro-mirror, the driving signal should have the same accuracy as the angle sensing signal. The circuit was designed with dual-channel and 12-bit digital-to-analog converters (DACs) (DAC8562); as shown in Figure 6, one channel was used to generate the drive control signal and the other was used as an automatic gain control signal.

The direct output signal of DAC is a ladder signal with high frequency noise and a mirror component, which leads to non-linear distortion of the waveform. In order to obtain pure waveform signals and filter these spurious components, a low-pass filter was added to the output of DAC, as shown in the Figure 7.

#### 3.1.3. Angle Sensing Signal Preprocessing

The output angle sensing current signal of the integrated angle sensing coil of the scanning grating micro-mirror passes through the current–voltage conversion circuit, as shown in Figure 8, and is then sampled by AD and sent into the FPGA. The amplitude information of the angle sensing signal is obtained by phase-locked amplification in the chip of the FPGA and the closed-loop control of the scanning grating micro-mirror is realized by the feedback control circuit of the amplitude control.

The principle of lock-in amplify is shown as follows: When there are two input signals, *U_s_* and *U_r_*, the amplitude of the output signal *U_o_* can be calculated by Equations (1)–(3):(1)$$U_{S}=E_{S}\sin(\omega t+\phi_{1})$$
(2)$$U_{r}=E_{r}\sin(\omega t+\phi_{2})$$
(3)$$U_{o}=U_{S}U_{r}=\frac{E_{S}E_{r}}{2}\left\{\cos\left(\phi_{1}-\phi_{2}\right)-\cos\left[2\omega t+\left(\phi_{1}-\phi_{2}\right)\right]\right\}$$
where *U_s_* is the angle sensor signal, *U_r_* is the reference signal, *E_s_* is the amplitude of *U_s_*, *E_r_* is the amplitude of *U_r_*, *ω* is the frequency of the drive signal and the angle sensor signal, *ϕ*_1_ is the phase of the angle sensor signal and *ϕ*_2_ is the phase of the reference signal.

The output signal *U_o_* includes a DC voltage and AC signal and *E_s_E_r_*(*cos*(*ϕ*_1_ − *ϕ*_2_)/2 is the DC part. *E_s_E_r_*{*cos*[(2*ωt* + (*ϕ*_1_ − *ϕ*_2_)]}/2 is the AC part, which has a frequency that is twice that of the drive signal, so it can be removed by the subsequent low-pass filter and the remaining DC part is proportional to the amplitude of the output signal of the angle sensor.

#### 3.1.4. Automatic Gain Circuit

In addition, in order to control the driving signal amplitude so that the scanning range of the MOEMS scanning grating micro-mirror can meet the requirements of the miniature near-infrared spectrometer, an AGC (automatic gain circuit) was used. In this paper, the automatic gain amplification of the driving signal was realized by AD603 of the Analog Devices (Norwood, MA, USA) amplifier in the programmable amplifier circuit mentioned above, as shown in Figure 9.

#### 3.1.5. PID Control Algorithm

The angular sensor signal sampled by ADC in FPGA enters the deviation into the digital PID controller. After the digital PID controller is processed, its output value is converted to an analog signal by D/A to control the controlled object. The usual expression of the digital PID control not only takes a long time to calculate but also takes up a lot of memory. Therefore, it is inconvenient to use Formula (4) to control directly:(4)$$u\left(k\right)=Kp\left\{e\left(k\right)+\frac{T}{T_{I}}\overset{n}{\underset{j=0}{\sum }}e\left(j\right)+\frac{T_{D}}{T}\left[e\left(k\right)-e\left(k-1\right)\right]\right\}$$
where *K* is sampling number, *Kp* is the proportionality coefficient, let $K_{I}=K_{p}\frac{T}{T_{i}}$ be the integral coefficient and $K_{D}=K_{p}\frac{T_{D}}{T}$ is the differential coefficient.

Therefore, the increment adopted in this paper was inconvenient. The output of the digital controller was only the increment of the control amount. The expression is shown in Formula (5):(5)$$\begin{array}{c} \Delta u=u\left(k\right)-u\left(k-1\right)\\ \;=K_{P}\left[e\left(k\right)-e\left(k-1\right)\right]+K_{I}\cdot e\left(k\right)+K_{D}[e\left(k\right)-2e\left(k-1\right)\\ \;+e\left(k-2\right)]\\ =K_{P}\Delta e\left(t\right)+K_{I}e\left(t\right)+K_{D}\left[\Delta e\left(k\right)-\Delta e\left(k-1\right)\right] \end{array}$$

Among them, the appropriate values of the corresponding proportional time constant, integral time constant and differential time constant were chosen according to the control requirement.

### 3.2. Signal Acquisition and Processing System

The acquisition circuit of the detector functions in a photovoltaic mode. In this mode, the InGaAs dual single detector diode and the photodiode do not need voltage; there is no dark current, so the output signal can be measured more accurately and the linearity of the output signal is better. The USB interface circuit was designed based on a Cypress 68013 series chip. The FPGA control circuit was designed based on an Altera Cyclone EP2C series chip.

#### 3.2.1. Detector Signal Acquisition

The output spectrum signal of the dual detectors is acquired and amplified through the instrument amplifier circuit with AD620 as the core, as shown in Figure 10. The reference voltage is connected to the non-inverting input of the AD620 amplifier of the Analog Devices (Norwood, MA, USA) and the reference voltage is set by adjusting the VR5 sliding resistor. The inverting input terminal is connected with the output of the amplitude detection circuit and the output signal of the AD620 can be expressed as *AD_2* = *K* × (*Vref* − *Detector_1*), where *Vref* is the differential signal equal to the dark current value of the detector 1 and *K* is obtained by adjusting the VR6 sliding resistor. The working principle of circuit b is the same as circuit a.

#### 3.2.2. ADC Module

In order to obtain a higher precision swing amplitude of the mirror and a higher acquisition rate of the spectrum signal, the ADC chip must have sufficient sample bits and a sufficient sampling rate. Figure 11 shows the ADC circuit of the signal acquisition module. AD9826 has three sampling channels and a 16-bits sample accuracy with up to a 15 Mbps sampling rate, which can meet the requirements of the spectrum and signal sampling. The output spectra and angular sensing signals of the two detectors are input into three channels of the ADC chip, respectively, to complete the analog-to-digital conversion.

#### 3.2.3. FPGA Main Control Module

The FPGA main control module is the core of the acquisition circuit, which realizes the sequence of the control and acquisition system. The control of the mirror and the module of the ADC and the USB interface need to work synchronously. Furthermore, according to the design requirements of the spectrometer resolution, the resonant frequency of the mirror is about 590 Hz and the swing is about 1.5 ms; the acquisition circuit needs to sample at least 6144 points of both spectrum signals and angle signals simultaneously, during the half-cycle period of about 0.89 ms, so the FPGA must work at a high speed and requires a huge data buffer space. Figure 12 shows the FPGA circuit designed with Altera Cyclone II EP2C5T144C8N FPGA chips.

#### 3.2.4. USB Interface Module

The collected spectrum data and angle signals are transmitted to the computer to be reconstructed. Due to the huge amount of information, the USB interface can perform high-speed data transmission. The designed USB 2.0 mode can transmit data at a rate of up to 480 Mbps. Figure 13 shows the USB interface circuit; the interface chip is CY7C68013.

## 4. Experiment and Results

### 4.1. Micro-Mirror Tests

As the core device of the micro-NIR spectrometer, the MOEMS scanning grating mirror must meet the requirements of equalizing the amplitude of swing and tilt angle vs. driving voltage as shown in Figure 14.

### 4.2. Micro-NIR Spectrometer Test

The internal structure of the spectrometer is shown in Figure 15a. The fiber and incident slit unit, the MOEMS integrated scanning grating micro-mirror unit and the detection system unit are located on the left side of the bottom plate and the collimating/focusing unit is located on the right side of the bottom plate. The circuit board was installed above the optical system. An internal partition was designed between the optical system and the circuit board. Three circular holes were designed on the internal partition to connect the MOEMS integrated scanning grating micro-mirror and detectors 1 and 2 with the circuit board, respectively. The side wall was designed with a USB interface, a power interface and a secondary development interface, as shown in Figure 15b.

The light source is introduced into the optical system through the optical fiber and then through the incident slit. The optical fiber and incident slit unit is composed of optical fiber, an incident slit, a clamp and a pedestal, as shown in Figure 16. One side of the flange circle is a slot and the other side is an SMA905 standard optical fiber interface.

The collimating/focusing lens unit is composed of a cover plate, a collimating/focusing lens and a pedestal. Figure 17 shows the assembly of the concave collimating/focusing lens and pedestal. The screw is fixed with the pedestal plate through the screw hole at the bottom of the pedestal and the arc on the other side can rotate around the screw to adjust the meridian angle deviation.

The MOEMS integrated scanning grating micro-mirror unit includes the MOEMS integrated scanning grating micro-mirror and pedestal, as shown in Figure 18. The circular arc at the bottom of the pedestal is similar to that of the collimating/focusing lens pedestal. The meridian angle deviation of the MOEMS integrated scanning grating micro-mirror can be adjusted by rotating the circular arc around the central axis.

The detection system unit includes a long-wave pass filter, a filter clamp, an exit slit, a detector, a detector clamp and a pedestal, as shown in Figure 19. Detector 1 (G10899; photosensitive surface diameter, 0.5 mm; response range, 0.5–1.7 μm) and detector 2 (G12183; photosensitive surface diameter, 0.5 mm; response range, 0.9–2.6 μm) are from Hamamatsu Photonics (Shizuoka Prefecture, Japan).

Figure 20 shows the micro-NIR spectrometer based on the MOEMS scanning grating mirror assembled with a system circuit PCB. The experimental test system was built based on the micro-mirror and the correct spectrum curve was displayed on the host computer.

When the MOEMS integrated scanning grating micro-mirror moves from −6 degrees to +6 degrees, the dual single-tube detector can complete one complete spectrum acquisition in the range of their respective response spectrum bands; detectors 1 and 2 can complete two complete spectrum acquisitions in the range of 800–2500 nm. The relationship between the scanning angle of the micro-mirror and the wavelength range of detectors is shown in Figure 21.

The experimental system was set up to test the performance of the system circuit and spectrometer. The output signal of the angle sensor and the original spectrum signal were sampled using an oscilloscope. The square wave driving signal was used to drive the MOEMS integrated scanning grating micro-mirror motion, as shown in Figure 22. At this point, the angle output of the original signal was about 2.5 mV in size, and the amplified signal was approximately 980 mV and the scanning angle was about 6.2 degrees.

The spectrum response curves of detectors 1 and 2 are shown in Figure 23.

Furthermore, the angle output signal and the spectrum signal collected by the detector 1 and 2 can also be transmitted to the upper computer and the upper computer reconstructed and spliced the spectrum signal through the angle sensor signal. Figure 24 shows the combined short wavelength and long wavelength spectrum.

Narrow-band filters were selected for the resolution test and the half-width of the filters was calculated. Figure 25 is the resolution test result of detector 1. Two narrow-band filters with a central wavelength of 1570 nm and 1580 nm were selected, respectively. It can be seen from the figure that the two characteristic peaks of 1570 nm and 1580 nm can be essentially separated. In general, the resolution of the central region of the spectrum detection range was better than that of the two ends of the spectrum detection range. Therefore, the spectrum resolution of detector 1 (800–1646 nm) was generally better than that of 12 nm.

Figure 26 is the resolution test result of detector 2. Narrow-band filters with a central wavelength of 2392 nm were selected for testing. The half-width of narrow-band filters with a central wavelength of 2392 nm was 17 nm (2399.2 − 2382.5 = 16.7 nm). Similarly, in general, the resolution of the central region of the spectrum detection range was better than that of the two ends of the spectrum detection range, so the spectrum resolution of detector 2 (1541–2532 nm) was better than that of 17 nm.

## 5. Conclusions

The results show that the control and acquisition system of the micro-NIR spectrometer based on the MOEMS scanning grating mirror allows the stable swing of the mirror and the effective signal acquisition of the spectrum signal and correct data transmission. The spectrum detection range of the NIR spectrometer was 800–2532 nm. The overall resolution of the spectrum was better than 12 nm.

## Figures and Tables

**Figure 1 micromachines-12-00696-f001:**
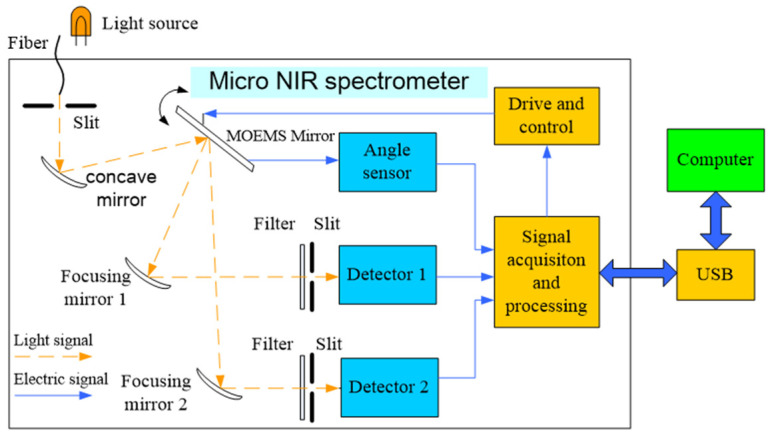
System structure principle of a micro-NIR spectrometer.

**Figure 2 micromachines-12-00696-f002:**
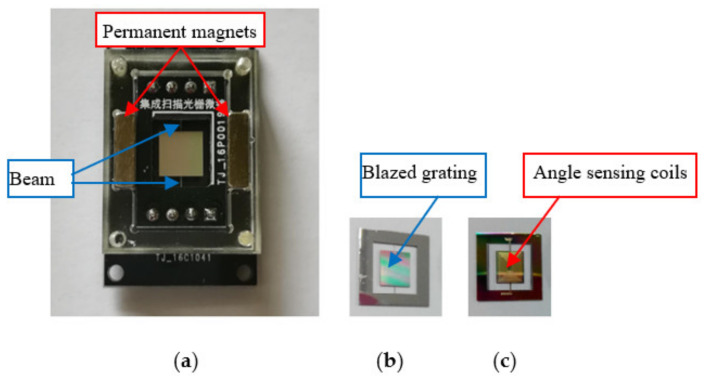
The MOEMS scanning grating mirror fixed on the PCB. (**a**) Fabricated scanning grating mirror. (**b**) Upper-side view. (**c**) Lower-side view.

**Figure 3 micromachines-12-00696-f003:**
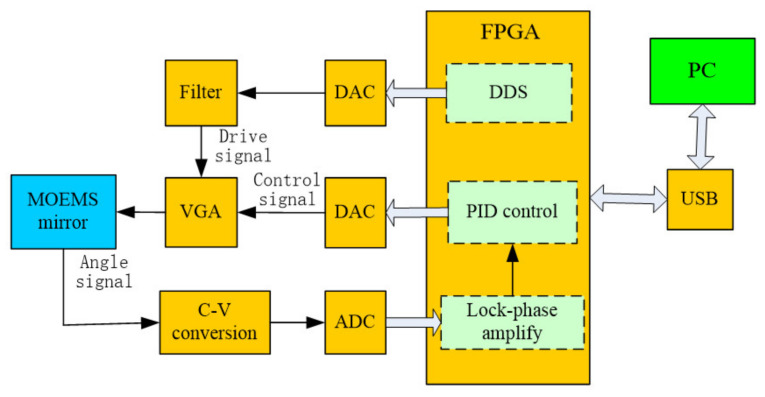
Structure block diagram of the drive and control system.

**Figure 4 micromachines-12-00696-f004:**
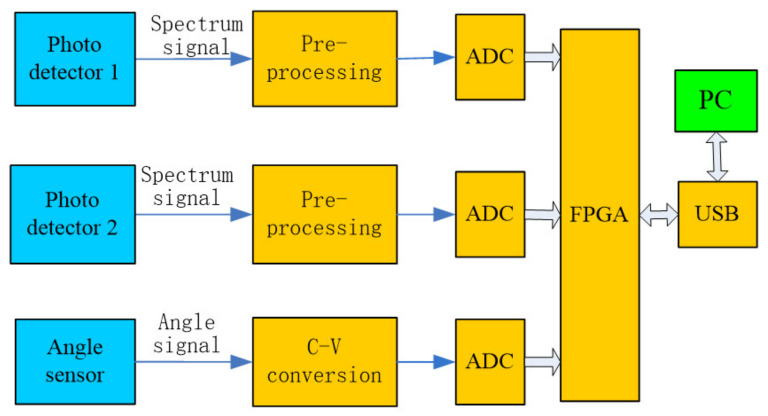
Structure block diagram of the signal acquisition system.

**Figure 5 micromachines-12-00696-f005:**
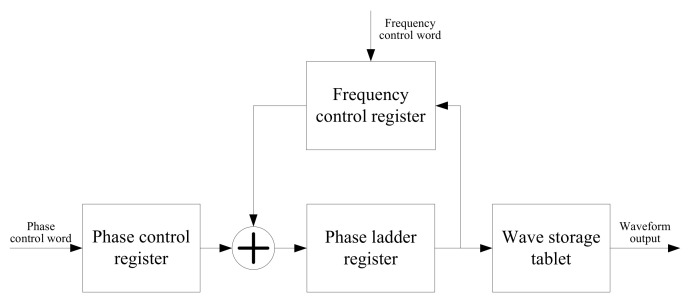
Schematic diagram of DDS.

**Figure 6 micromachines-12-00696-f006:**
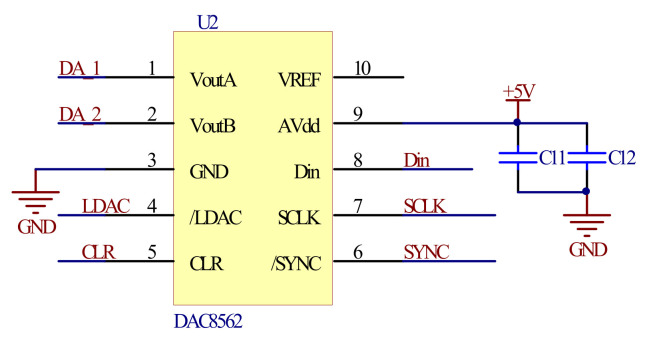
D/A converter circuit.

**Figure 7 micromachines-12-00696-f007:**
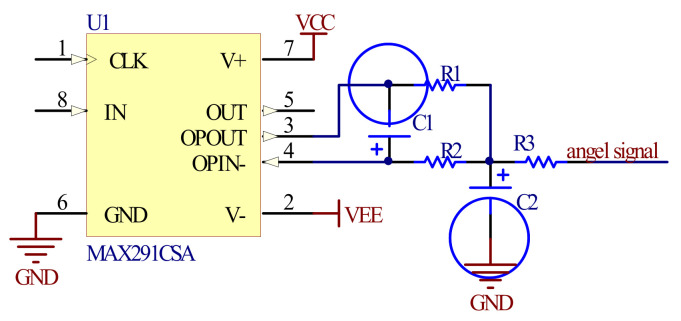
The second-order Butterworth low-pass filter.

**Figure 8 micromachines-12-00696-f008:**
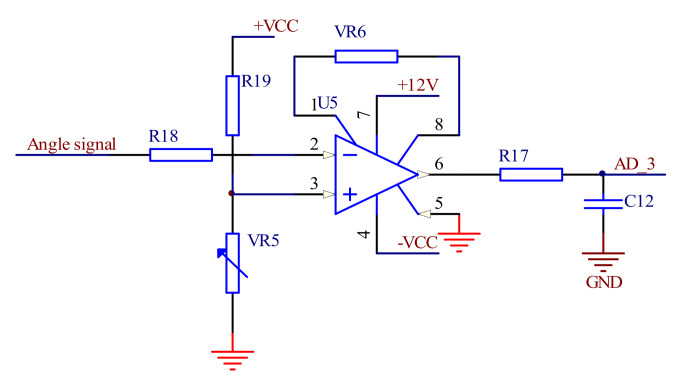
Angle sensing signal preprocessing circuit.

**Figure 9 micromachines-12-00696-f009:**
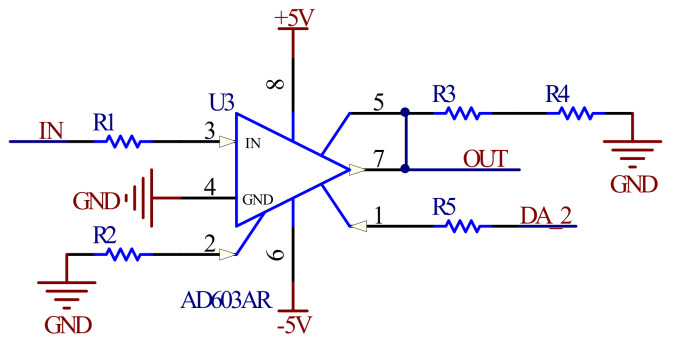
Automatic gain circuit.

**Figure 10 micromachines-12-00696-f010:**
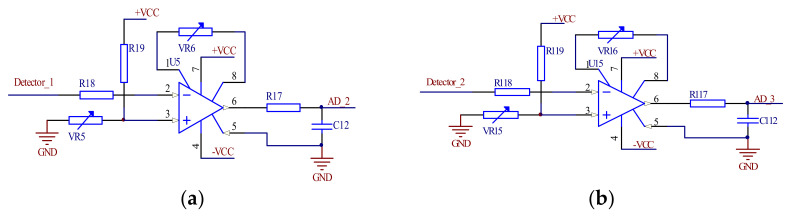
Double detector signal acquisition circuit. (**a**) Circuit of detector 1. (**b**) Circuit of detector 2.

**Figure 11 micromachines-12-00696-f011:**
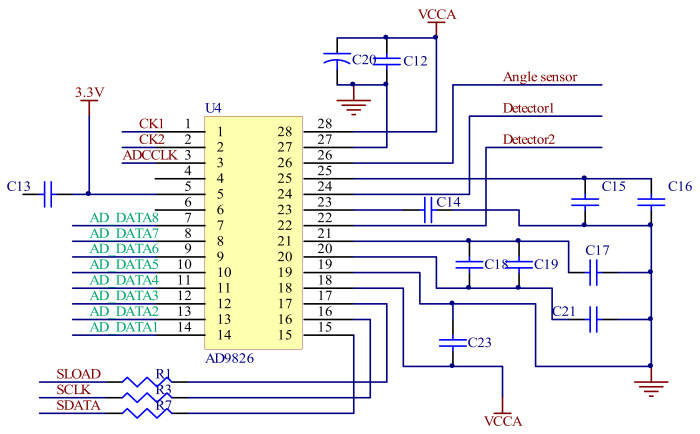
ADC circuit of the signal acquisition system.

**Figure 12 micromachines-12-00696-f012:**
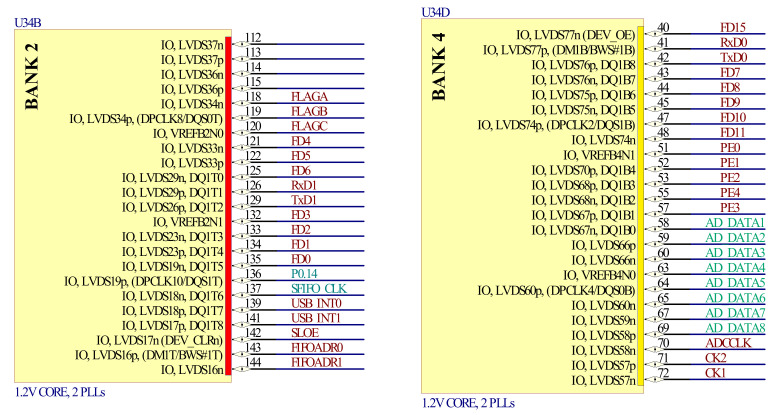
FPGA main control circuit.

**Figure 13 micromachines-12-00696-f013:**
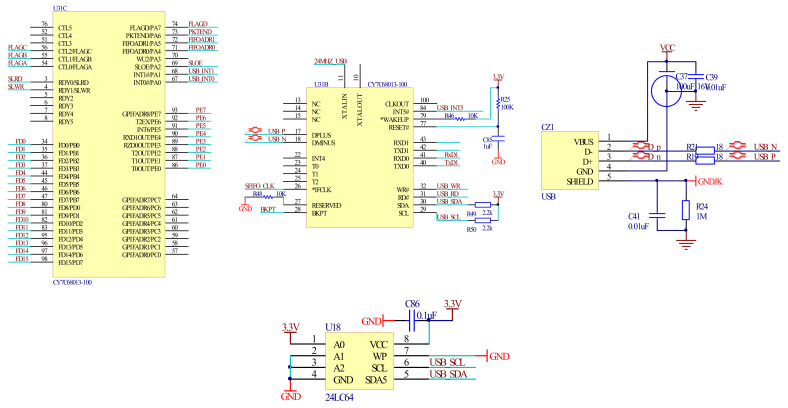
USB interface circuit.

**Figure 14 micromachines-12-00696-f014:**
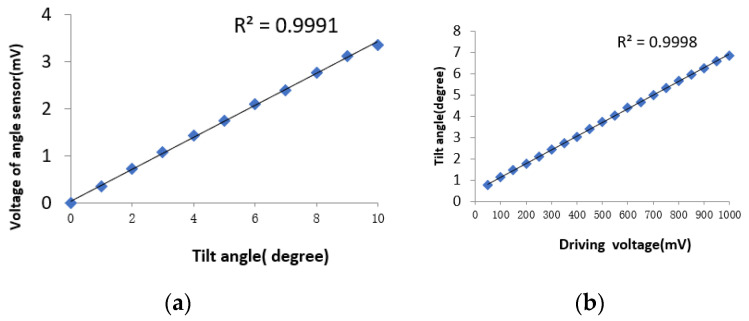
The test of the MOEMS scanning grating mirror. (**a**) Voltage of angle sensor vs. tilt angle. (**b**) Tilt angle vs. driving voltage.

**Figure 15 micromachines-12-00696-f015:**
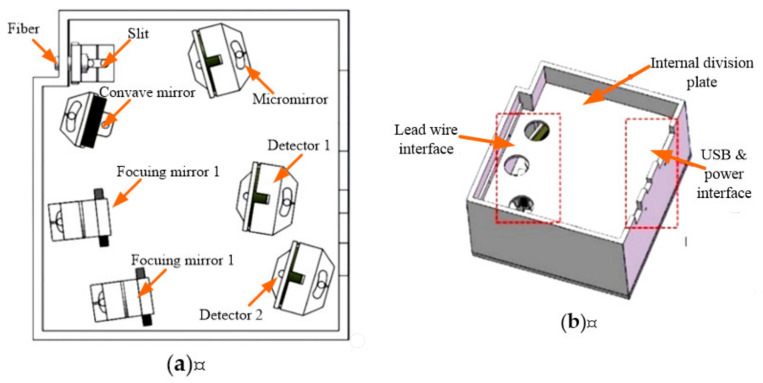
The layout of the designed micro-NIR spectrometer. (**a**) Internal structure layout. (**b**) Internal partition between the optical system and circuit board.

**Figure 16 micromachines-12-00696-f016:**
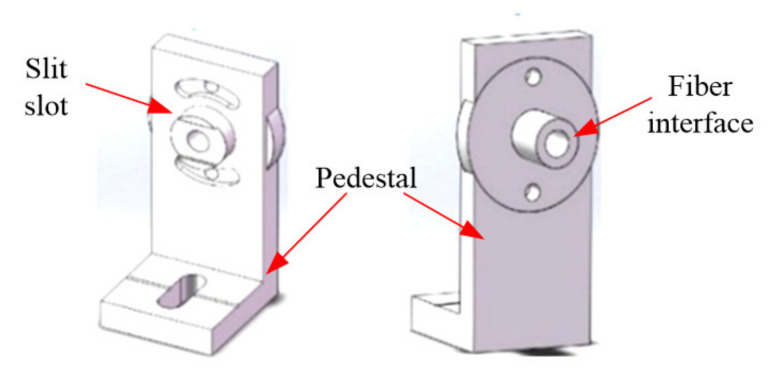
The components of the optical fiber and entrance slit.

**Figure 17 micromachines-12-00696-f017:**
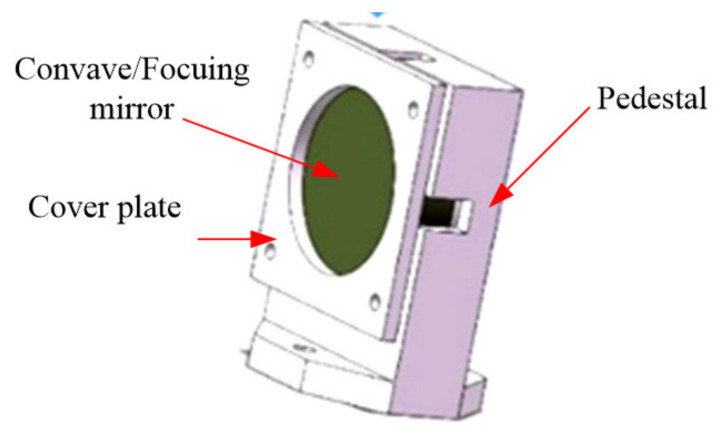
The components of the concave reflecting mirror.

**Figure 18 micromachines-12-00696-f018:**
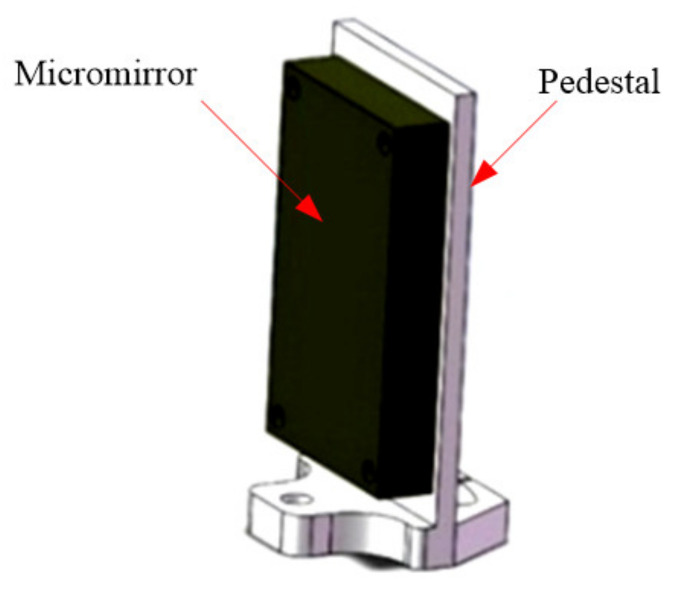
The components of the MOEMS scanning grating mirror.

**Figure 19 micromachines-12-00696-f019:**
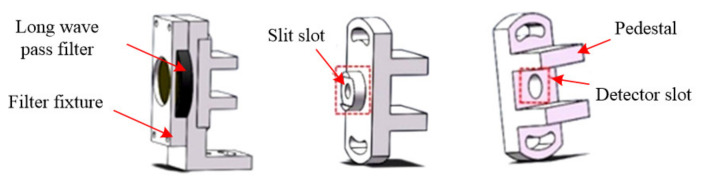
The components of the detection system.

**Figure 20 micromachines-12-00696-f020:**
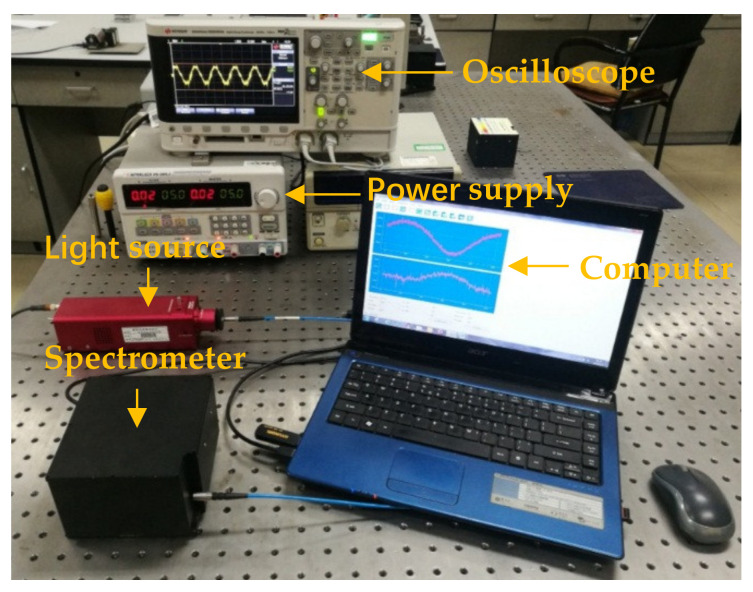
Photo of the assembled micro NIR spectrometer.

**Figure 21 micromachines-12-00696-f021:**
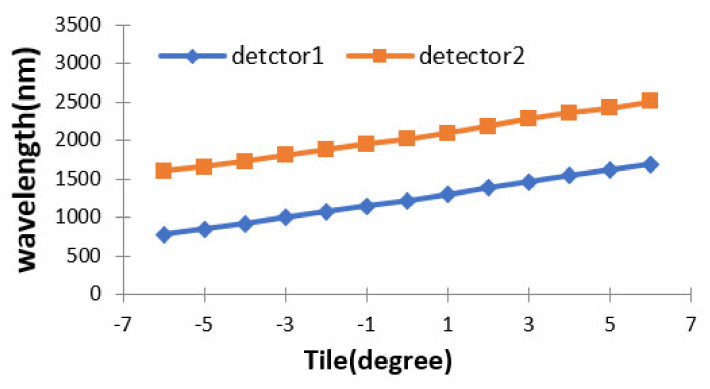
The relationship between scanning angle and wavelength range of the detectors.

**Figure 22 micromachines-12-00696-f022:**
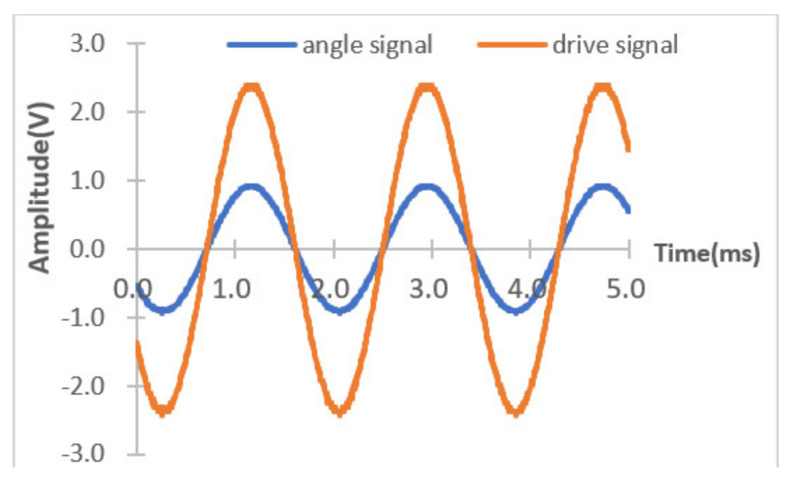
Driving signal and output signal of the MOEMS mirror.

**Figure 23 micromachines-12-00696-f023:**
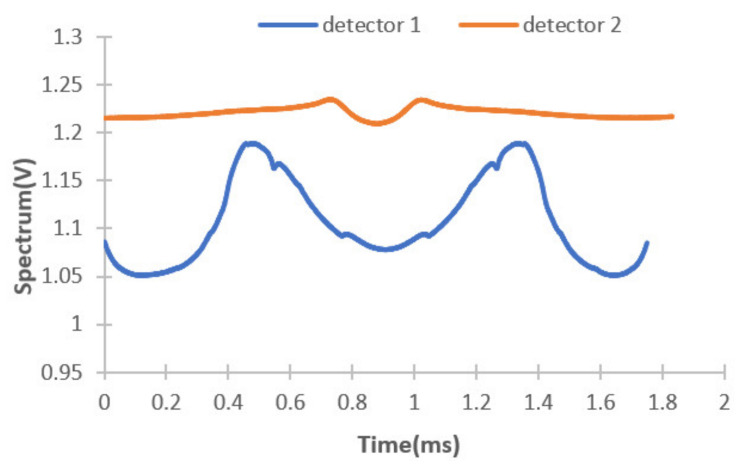
Spectrum signal of detector 1 and detector 2.

**Figure 24 micromachines-12-00696-f024:**
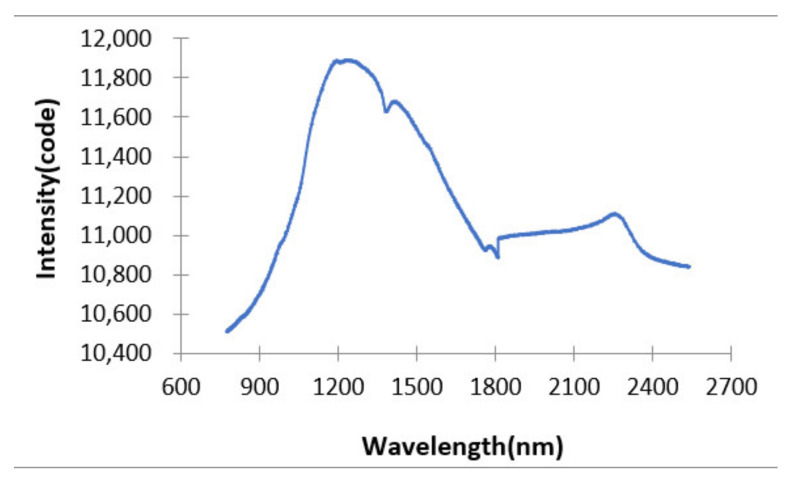
The combined short wavelength and long wavelength spectra.

**Figure 25 micromachines-12-00696-f025:**
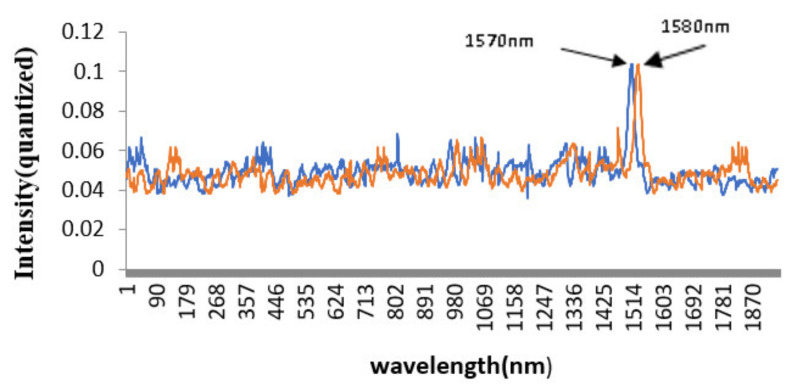
The testing of the resolution of detector 1.

**Figure 26 micromachines-12-00696-f026:**
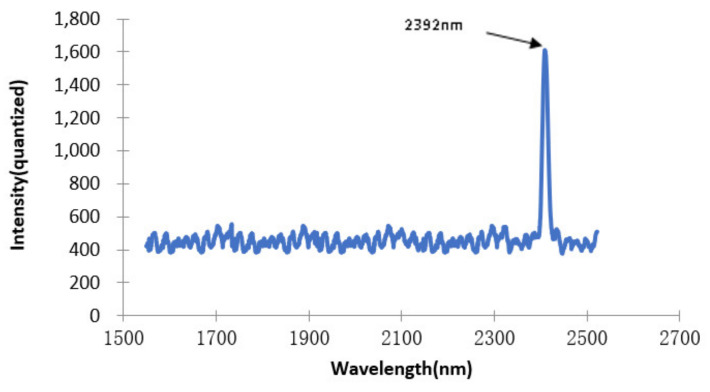
The testing of the resolution of detector 2.

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
