# Peer review of "Control and Signal Acquisition System of Broad-Spectrum Micro-Near-Infrared Spectrometer Based on Dual Single Detector"

_micromachines, 2021, doi:10.3390/mi12060696_

Round 1

Reviewer 1 Report

A well written paper which presents an interesting methodology. Here a few comments for the authors -

  1. The results presented are only in the linear operating regime, it will be interesting to see what happens when driving voltage is increased to a point where non-linearity starts to dominate; does that have any effect on the control system?
  2. Assembled NIR spectrometer and test setup should be shown in separate figures for elaboration in figure 15.
  3. Typo on X axis label on fig 16
  4. It will be more presentable if the actual waveforms were saved on the oscilloscope and then plotted using Matlab, Excel etc.. It is hard for the readers to focus in on these images (figure 17-18)
  5. This can be out of scope for the study but have any sample devices been tested under temperature cycling to see if hysteresis can be an issue?

Author Response

Dear Reviewer:

Thank you very much for you recommendation! I will reply all the comments and suggestions one by one:

  1. The results presented are only in the linear operating regime, it will be interesting to see what happens when driving voltage is increased to a point where non-linearity starts to dominate; does that have any effect on the control system?

Reply: when driving voltage is increased to a point where non-linearity starts to dominate,there is no problem for the circuit, but for the micromirror, when the swing angle of the micromirror exceeds the range of good linearity, the performance of the corresponding spectrum will become very poor, so the appropriate driving voltage is selected to control the swing of the micromirror.
2. Assembled NIR spectrometer and test setup should be shown in separate figures for elaboration in figure 15.

Reply: Figure 15 is marked in detail, and the internal structure of the spectrometer is described in detail3. Typo on X axis label on fig 16

Reply: X axis label on fig 16 have been typed.
4. It will be more presentable if the actual waveforms were saved on the oscilloscope and then plotted using Matlab,Excel etc.. It is hard for the readers to focus in on these images (figure 17-18)

Reply:The oscilloscopes are used to save the data and then draw the graphs in Excel, which are now very clear and readable

  1. This can be out of scope for the study but have any sample devices been tested under temperature cycling to see if hysteresis can be an issue?

Reply: For the current development of the device, for the conventional environment, to achieve high-precision control, the next step will be to carry out special high and low temperature change experiment.

Reviewer 2 Report

  • The paper is very interesting and innovative given that I have some background knowledge on IR micro-spectrometer design
  • Sentence structure must be improved, see my comments in the attached file
  • Expand conclusion with more remarks on what can be improved and limitations of your setup
  • Provide more details, specifications on the spectrometer optical design such as entrance and exit slit width
  • Show more results on the performance of the automatic gain circuit i.e. did you manage to control the micro-scanner amplitude variation due to temperature effects?
  • Mention more details about the detectors selected. Are they the same? I am asking this because there is a sharp change in the spectral signal between one detector and the other.

Author Response

Dear reviewer:

Thank you very much for you recommendation! I will reply all the comments and suggestions one by one:

The paper is very interesting and innovative given that I have some background knowledge on IR micro-spectrometer design Sentence structure must be improved, see my comments in the attached file
Expand conclusion with more remarks on what can be improved and limitations of your setup

Reply: we have rewritten the entire article, and at the same time, I have asked my English colleagues to finish polishing the article.
Provide more details, specifications on the spectrometer optical design such as entrance and exit slit width

Reply:About the spectrometer optical design, The structure design of the spectrometer has been added in detail.

Show more results on the performance of the automatic gain circuit i.e. did you manage to control the micro-scanner amplitude variation due to temperature effects?

Reply: In this paper, the structure design of the spectrometer has been added in detail. The automatic gain circuit can automatically adjust the gain by detecting the size of the output angle sensing signal, generating the appropriate driving voltage and controlling the amplification factor of the controllable gain amplifier.
Mention more details about the detectors selected. Are they the same? I am asking this because there is a sharp change in the spectral signal between one detector and the other.

Reply:The models and specifications of the detectors have been listed in the paper, which are two kinds of InGaAs detectors with different specifications,the reason for there is a sharp change may be caused by the structure of the light path.